# Numerical Simulation of Swirling Impinging Jet Issuing from a Threaded Hole under Inclined Condition

**DOI:** 10.3390/e22010015

**Published:** 2019-12-22

**Authors:** Liang Xu, Yanh Xiong, Lei Xi, Jianmin Gao, Yunlong Li, Zhen Zhao

**Affiliations:** 1State key Laboratory of Mechanical Manufacturing Systems Engineering, Xi’an Jiaotong University, Xi’an 710049, China; xuliang@mail.xjtu.edu.cn (L.X.); xileibisheng@stu.xjtu.edu.cn (L.X.); gjm@mail.xjtu.edu.cn (J.G.); ylongli@mail.xjtu.edu.cn (Y.L.);; 2Guangdong Xi’an Jiaotong University Academy, Foshan 528000, China

**Keywords:** swirling impinging jet, inclination angle, heat transfer characteristic, numerical simulation

## Abstract

There are some inclined jet holes in the cooling structure of the leading edge region of gas turbine blades. In order to improve the cooling effect of traditional round holes, this paper proposes to replace the round holes with threaded holes, and studies the complex flow and heat transfer performance of the swirling impinging jet (SIJ) issuing from the 45° threaded holes in the inclined condition by numerical simulation. The influencing factors include jet inclination angle *α* (45°–90°), jet-to-plate distance (*H/d* = 2, 4, 6), and Reynolds number (6000–24,000). The results show that the inclination angle and jet-to-plate distance have a great influence on the size, shape, and position of vortices in the jet space, while the Reynolds number has little effect on the vortices. In the inclined state, the impinging cooling effect of the swirling impinging jet is better than that of the circular impinging jet (CIJ), both heat transfer coefficients will degrade significantly when the inclination angle is 45°. When the inclination angle is greater than 45°, compared with the round hole, the enhanced heat transfer region for the swirling jet is in the region of r/d < 3, while both of the *Nusselt* numbers in the wall jet region are weak, with a value of just 20. At the same time, with the increasing of the inclination angle (α > 45°), the average *Nusselt* number on target surface holds a constant value. Under the inclined conditions, the heat transfer coefficient on the target surface for the swirling jet is increased totally with the increasing of the *Re*, but when the *Re* is larger than 18,000, the rate of enhanced heat transfer gradually weakens.

## 1. Introduction

As one of the most effective convective heat transfer enhancement technologies using a single-phase cooling media, the impinging jet is widely used in gas turbine cooling blades, electronic components, textiles, paper drying, metallurgical industry, and so on. In some practical applications, owing to space constraints, jet holes have to be arranged obliquely, such as the layout of the impact cooling holes on the leading edge of turbine blades, and the cooling performance of the impinging jet under the inclined condition is sure to be affected significantly and needs to be further researched.

In the aspect of the inclined impingement jet, Yoon [1], Akansu [2], and Beitelmal [3] studied the influence of the inclined impingement jet for a single round hole on the heat transfer characteristics of a heating plate. They found that, with the decrease of inclination angle, the stagnation point moves to the micro-flow area, and the local *Nu* number in the micro-flow area is larger than that in the large-flow area. Yan [4], Sparrow [5], and O’Donovan [6] studied the circular impinging jet (CIJ) under inclined conditions. Their results showed the smaller the impinging angle of the jets, the farther the *Nu* peak point is away from the center of the target surface. Goldstein et al. [7] measured the local temperature and heat flux density of the plate surface. They pointed out that the average *Nu* number of vertical jet near the peak area (0 < r/d < 1) was the highest when the jet distance and the jet impact angle were held at a constant value. An experimental study by Yi [8] showed that a larger inclination angle can improve the cooling performance when the jet-to-plate distance is constant, and a larger jet-to-plate distance can reduce the cooling performance when the inclination angle is constant. Bhagwat et al. [9] used the *V2F* model to study the CIJ under inclined conditions. The results showed that *Nu* number at the upper end of the plate decreases sharply when the plate is inclined, while *Nu* number decreases slowly at the lower part of the plate. Chuang [10] and Chen [11] used the *k-ε* model to analyze the inclined impinging jet. Their results also showed that, when the inclination angle decreases, the left recirculation area decreases, the right recirculation area increases, and the position of the maximum pressure area on the target surface moves to the left. Florian et al. [12] analyzed impinging jet cooling with a large eddy simulation. The results show that the position of the stagnation point is different from that of the maximum Nusselt number when the plate is inclined, and the heat transfer can be enhanced by increasing the Reynolds number. On the basis of the analysis of the above literature, the impinging jet with oblique incidence can cause the maximum heat transfer point to move upstream of the target surface, the heat transfer performance for the oblique impinging jet is deteriorated significantly compared with the vertical impinging jet, and it is necessary to design new nozzles with some special structures to change the adverse cooling situation when the inclination impinging jet happens.

In recent years, many scholars have found that the swirling impinging jet (SIJ) is much better than the conventional circular impinging jet in improving the heat transfer performance of the target surface [13,14,15,16,17]. Although the use of SIJ for cooling turbine blades will make the blade structure more complex and difficult to process, with the development of metal 3D printing technology, it is possible to design and manufacture turbine blades with a complex cooling structure [18,19]. For the swirling impinging jet, Bakirci et al. [20] found that increasing the helical angle can increase the uniformity of radial heat transfer on the plate surface. Ahmed et al. [21,22,23,24] studied the impingement pressure, flow, and heat transfer characteristics of swirling and non-swirling flows. The results showed that the pressure coefficient at the stagnation point is the same as that of non-swirling flow for the low-intensity swirling flow, and has a maximum value. For the medium-intensity to high-intensity swirling flow, the maximum pressure coefficient points move outward along the radial direction. The total convective heat transfer coefficient can be increased by increasing swirl intensity at a lower jet-to-plate distance, and the non-swirl cooling effect is better at a long impinging distance. Burak [25] experimentally studied the heat transfer characteristics and pressure distribution of a vortex coaxial confinement impinging jet. Amini [26] and Nanan [27] studied the SIJ formed by twisting bands. They also showed that the swirling jet made the heat transfer more uniform on the plate, decreased the coefficient of heat transfer a little near the stagnation point, and enhanced the heat transfer coefficient at the wall jet outward region. Borynyak et al. [28] used a large eddy simulation to study the low swirling impinging jet with a convergent nozzle. They found that the axial velocity of swirl jet under turbulent stress increased in the near region of the jet compared with that of non-swirl jet. Ortega [29] numerically studied the effects of Reynolds number, swirl intensity, average turbulence intensity of jet, and jet-to-plate distance on heat transfer characteristics on the plate surface. They also studied the effects of eddy parameters (eddy current intensity S and eddy core δ) on heat transfer on the heated plate surface, and optimized the area-weighted average Nusselt number, Nusselt number at the stagnation point, and heat transfer uniformity. Illyas et al. [30] experimentally studied the flow characteristics of SIJ with different helical surfaces. Nuntadusit et al. [31] studied the flow and heat transfer characteristics of porous swirling impinging jets (SIJS) and circular hole jets (CIJS). Drost et al. [32] studied a numerical procedure for the prediction of local entropy generation rates and the application of that procedure to convective heat transfer associated with a fluid jet impinging on a heated wall. Systems of individual components can be optimized from estimates of net entropy generation for complete components, but the development of novel components and processes should benefit from knowing the distribution and sources of entropy generation on a local level. In conclusion, to achieve uniform cooling at high heat transfer rates, the swirling impinging jet is a promising method. At present, there are no specific studies on flow and heat transfer in an inclined SIJ issuing from some special nozzles. So, the future focus of inclined impingement cooling technology may be to develop some interesting special holes to generate the swirling flow, which can be manufactured by 3D printing. 

In the literature [14], we studied the swirling impinging jet of thread-like holes and pointed out that the SIJ of the thread-like hole with a helix angle of 45° causes the target surface to have a better convective heat transfer effect. The swirler consists of a smooth circular hole and four circumferential screw grooves for structuring the impinging jet flow with a superimposed swirl, so that the nozzle is similar to the conventional circular hole with internal threads. Supported by 3D printing technology, the traditional round holes are easily replaced with these thread-like holes. In the literature [16], the heat transfer characteristics of swirling impinging jet with different helix angles (0°–75°) were studied by our research group. For the vertical impinging jet, when *H/d* = 2 and *Re* = 4000, increasing the helix angle makes the Nusselt number in the central region of the plate increase, but the heat transfer uniformity gradually becomes worse. The heat transfer effect with a helix angle of 45° is slightly lower than that with a helix angle of 60°. When the impact distance is *H/d* = 4 and *Re* = 6000, the heat transfer effect of the plate surface is the best when the helix angle is 45°, while the heat transfer effect of the 75° is close to that of the round hole. Therefore, in this paper, the helical angle of 45° is chosen as a representative to study the flow and heat transfer characteristics of the swirling impinging jet under the inclined conditions. The purpose is to explore the enhanced performance using the swirling jet issuing from the 45° threaded hole under the inclined impinging jet. Firstly, according to the experimental datum from the vertical swirling impinging jet in the literature [14], and the inclined circular impinging jet in the work of [6], a variety of turbulence models are used to verify the calculation method suitable for calculating the SIJ under the inclined condition. Then, by changing the angle of inclination, jet-to-plate distance, Reynolds number, and other parameters, the flow and heat transfer characteristics of the CIJ and the SIJ are compared and analyzed in detail.

## 2. Numerical Calculation Method

In this paper, the thread-like hole with a helical angle of 45° in the work of [14] is selected to form swirling flow, as showed in Figure 1a. According to the authors of [14], the nondimensional swirl number (*S*) of the swirling jet at the nozzle exit is about 0.18, and the excited flow belongs to a low-intensity swirling flow. The calculation model is shown in Figure 1b. The inclination angle is α, and the angle of the vertical jet is 90°. Under the inclined conditions, because the temperature distribution on the target surface of the SIJ is not uniform, in order to keep the data consistent, the linear *r* (azimuth angle *β* between the line *r* and the X axis) and the plane Z_0_ (X0Y plane) on the target surface are selected for data analysis, and their positions are shown in Figure 1b.

In the calculation model, the length of the inlet pipe is 10 d to ensure the full development of air flow in the pipe. The length × width × height of the three-dimensional model are 300 × 300 × 110 (mm). ICEM CFD is used to draw the multi-block structured grids. In order to capture the flow field near the wall accurately, the value of the y+ function is generally recommended to be less than 1 when the SST *k-w* turbulence model is selected. Thus, the interface region of the jet space above the target surface is chosen as a focus domain to use a structured and refined grid. In this paper, the size of the first layer near the wall is 0.001 mm with the grid growth rate of 1.2, and then the value of the y+ function is about 0.8. Figure 2a shows the grid model. The computational grid is considered grid-independent when the *Nu* of the target surface varies within 2%, as shown in Figure 2b. The calculation software in this paper is commercial software ANSYS FLUENT 18.0. The SST *k-w* turbulence model and pressure-based solver are selected. The second-order upwind scheme is selected for the calculation method. The absolute temperature of 300 K, the turbulent intensity of 5%, and the mass flow rate (*m* = *Reρdμ*/4, where *m* is the mass flow rate and *Re* is Reynolds number, *Re* = *ρU*_0_*d*/*μ*. *U*_0_ is the mean axial velocity; *d* is the equivalent diameter of 9 mm; and *ρ* and *μ* are the density and viscosity of air at the jet exit, respectively) based on a given *Re* were specified at the inlet boundary condition. The range of *Re* was varied from 6000 to 24,000. The outlet boundary pressure is specified as the atmospheric pressure. The target is modeled as a stainless steel solid, the bottom target surface is set as a constant heat flux of 1000 W/m^2^, the upper surface of the target is given as the fluid–solid coupling interface, and four edges of the target are modeled as adiabatic walls. The nozzle walls were modeled as adiabatic walls with a non-slip boundary.

The data processing methods in this paper are as follows:

Local heat transfer coefficient:(1)$$c=q/(T_{w}-T_{airinlet}).$$

Reynolds number is defined as follows:(2)$$Re=\rho u_{0}d/\mu =4m/\pi du_{0}.$$

Nusselt number is defined as follows
(3)Nu=cd/k.

The average Nusselt (*Nu_ave_*) number:(4)$$Nu_{ave}=\frac{1}{A}\iint NudA=\frac{1}{\pi r^{2}}\int_{0}^{r}\int_{0}^{2\pi }Nu\cdot xd\theta dx=\frac{2}{r^{2}}\int_{0}^{r}Nu\cdot xdx,$$
where *q* is the target surface heat flux, W/m^2^; *T_w_* is local target surface temperature; *T_airinlet_* is the inlet air temperature, 300 K; *d* is the inlet hole diameter, 9 mm; *k* is the air thermal conductivity, W/m/k; *μ* is the aerodynamic viscosity; and *m* is the mass flow rate.

In order to verify the reliability of the calculation model in this paper, the vertical jet experimental data in the works of [14] and the experimental data of different inclination angles (60°, 75°, 90°) in the literature [6] are validated. Figure 3 shows the Nu value of the target surface calculated by different turbulence models. As can be seen from Figure 3a, the stagnation region of vertical SIJ (*r/d* < 3) is a region of laminar flow and transitional development. The turbulence model is used to simulate the stagnation region, which results in a high calculation error (up to 30%). For the circular impinging jet, the turbulent flow simulation results make it difficult to capture the fluctuation of the second peak on the target surface, which is consistent with other numerical calculations [9]. In terms of the coincidence with the experimental results, the calculation results using the *SST k-ω* model are the closest to the experimental datum of the above literature, so it is used to calculate the swirling impinging jet under the inclined conditions.

## 3. Results and Analysis

In order to study the flow and heat transfer performance of the 45° threaded hole impinging jet under the inclined conditions, this paper mainly studies the factors of inclined angle *α*, impinging distance *H*, and Reynolds number *Re* of the inlet jet, and compares them with the traditional circular impinging jet. The selected operating parameters are shown in Table 1. Table 2 shows the mass flow rates and pressure corresponding to different Reynolds numbers.

### 3.1. Flow Field

Figure 4 shows the streamlines and velocity distribution of circular and swirling jet in the Z_0_ plane at different inclination angles at *Re* = 6000 and *H/d* = 2. The left side of the nozzle is a traditional smooth circular hole and the right side is the 45° threaded hole. When the vertical jet (*α* = 90°) is applied, the flow field and velocity field are symmetrical in both the circular and swirling jets. In Figure 4a, it is obvious that the high-speed impinging jet from a smooth circular hole entrains the surrounding air. The entraining air and the air flowing out of the jet space after the wall jet form a large recirculation vortex, and the center of the vortex is located at *r/d* = ±13.5. In the case of the vertical SIJ, because the swirling jet has a stronger entrainment effect on the external air flow, it can be seen that the position of the vortex center is closer to the stagnation point of the jet than that of the smooth circular hole jet, which appears at *r/d* = ±12. At the same time, owing to the blocking effect of the internal threads, most of the airflow is ejected from the central region of the threaded holes. Compared with the smooth circular holes, the jet velocity at the central region of the exit for the swirling jet has a bigger value and the potential flow develops further in jet space, shown as the red region in Figure 4a, which will enhance the transfer effect of the jet stagnation zone. 

When *α* is equal to 75°, for the circular hole jet, more air flows downstream of the plate, and the entrainment effect of the jet on the upstream direction of the plate is enhanced, which causes the upstream direction wall jet to occur earlier than that of the vertical jet, so the center of the recirculation vortex is slightly close to the stagnation point, and the expansion area of the vortex is also increased. For an SIJ, the recirculation vortex upstream of the target also moves towards the stagnation point of the jet by a stronger entrainment of potential flow and the location of its center is about *r/d* = −7.5. Compared with the circular hole jet, a new entrainment vortex adjacent to the outlet boundary of jet space is formed at *r/d* = −15. This may be that there are mixeds region of two stream flows, one is the wall jet flow out of jet space with a tangential velocity component and the other is the surrounding airflow entrained by the circumfluence vortex. In the downstream region, the wall jet occurs closer to the stagnation point, the central of the recirculation vortex moves ahead to the location of about *r/d* = 9, and a half-baked entrainment vortex also appears at the exit of the jet space. Meanwhile, the wall boundary layer thickness of the airflow near the downstream region of the wall jet increases correspondingly, which will weaken the heat transfer performance of the wall region.

When *α* is equal to 60°, the center of the circumfluence vortex in the upstream direction of the CIJ moves to about *r/d* = −7. At this time, the range of the vortex is further expanded, but there is still a small amount of airflow escaping from the upstream direction near the target surface. In the downstream direction of the target, most of the airflow flows out from the downstream region and the suction effect on the surrounding airflow of the downstream outlet boundary is very weak. The flow with radial velocity as the dominant factor has a certain entrainment effect on the upper low-speed flow in the jet space. In the region of about *r/d* ≥ 8, the disturbance flow caused by the entrainment effect can be clearly seen, and a recirculation vortex with an obscure central occurs at about *r/d* = 10. For the SIJ, because of its strong entrainment effect on the external air flow in the upstream region, the rebounding airflow in the stagnation region is suppressed, and then a small area of wall-like corner vortex is formed at *r/d* = −4. At this time, there is no jet escaping from the upstream area. All the jets escape out of the jet space from the downstream, and the velocity of the airflow near the downstream target surface increases, which has a strong entrainment effect on the upper airflow in the jet space. It can be seen that an obvious entrainment vortex is formed at *r/d* = 7.5. With the gradual weakening of the outlet airflow, the entrainment effect is gradually weakened, and the height of vortex in the jet space is gradually reduced.

When *α* is equal to 45°, all the airflow of the CIJ escapes out of the jet space from downstream of the target surface. A low-speed large recirculation vortex occupies almost the whole jet space in the upstream region and is suppressed the upstream wall jet of the rebound jet in the stagnation zone, which will seriously deteriorate the heat transfer performance in the upstream region. In the downstream region, the airflow increases obviously, and the wall jet moves outward. The entrained airflow is not seen until the region of *r/d* ≥ 12.5. For the SIJ, the entrainment airflow mostly comes from the upstream region, which suppresses the rebound airflow in the stagnant zone to develop along the upstream region. A very small wall-like corner vortex is formed near the stagnation point about *r/d* = 2. At the same time, the location of the wall jet also moves out in the downstream region of the target surface, but, owing to the strong entrainment effect of the airflow near the wall, a distinct entrainment vortex is formed in the exit region, and the center of the vortex is about *r/d* = 15.

In conclusion, for the smooth circular impinging jet, reducing the inclination angle causes the recirculation vortices in the upstream direction to move toward the stagnation point, and the intensity of the recirculation vortex increases gradually; even the low-speed recirculation flow occupies the whole upstream jet space. For an SIJ, with the decrease in the inclination angle, the recirculation vortices of upstream jet space gradually weaken; the rebound airflow in the stagnant zone is suppressed by more and more the entrained airflow from the upstream region, and eventually becomes a wall-like corner vortex near stagnation point. When the circular hole jet is inclined to 45°, all the impinging jets escape out of jet space from the downstream region, while that situation happens at *α* = 60° for the SIJ without a large recirculation vortex in the upstream region. Even when all the airflow escapes out of jet space from the downstream region, there is a smooth suction airflow in the upstream region for the SIJ, thereby improving heat transfer performance in the upstream region. Secondly, by comparing the velocity contours of swirling jet and circular hole jet, it can be clearly found the flow velocity at the hole exit of the SIJ is obviously higher than that of the CIJ owing to the use of threaded holes, which makes a higher convective heat transfer capacity near the stagnation zone.

Figure 5 shows the streamlines and velocity contours of CIJ and SIJ on the Z_0_ plane with different jet-to-plate distances at *Re* = 6000 and *α* = 45°. When *H/d* is equal to 4, compared with *H/d* = 2 (Figure 4d), the surrounding airflow entrained by the potential flow mostly comes from the upstream region of the jet space, and the rebound airflow in the stagnation zone is restrained; thus, a wall-like corner vortex is formed at *r/d* = −2.5. With the increase in the jet-to-plate distance (*H/d* = 6), the corner vortex is further suppressed, and the extended region is greatly reduced. In the downstream, owing to the entrainment of airflow near the wall, an obvious entrainment vortex is formed in the exit region, and the center of the vortex is close to the exit region (r/d ≥ 15). For the inclined SIJ with a large jet-to-plate distance (*H/d* = 4), the airflow sucked into the upstream direction can also flow smoothly from the outside surrounding airflow. The wall jet of the rebound air flow in the stagnation zone is also suppressed by these suction airflows and forms a wall angle-like vortex near the stagnation point. Compared with the CIJ under the same working conditions, the expansion range of the vortex is smaller and closer to the stagnation point. When *H/d* is equal to 6, the recirculation vortex disappears, but when the suction airflow near the wall in upstream region meets with the rebound jet in the stagnation zone, a very small vortex is formed at *r/d* = −4. In the downstream region, because the downstream airflow contains tangential velocity components, and there are double effects of shear and entrainment on the airflow near the wall, the downstream airflow is very disordered without the vortex.

Figure 6 shows the streamlines and velocity contours of CIJ and SIJ with different Reynolds numbers at *α* = 45° and *H/d* = 2. When the Reynolds number increases from 6000 to 24,000, the suction effect of the jet on the external airflow increases gradually with the increase of the velocity of the impinging jet. The more the airflow in the upstream region is entrained by the CIJ, the stronger the suppression effect on the recirculation vortex and the smaller the scale of the upstream vortex. However, the radial position of the center of the vortex is basically kept at *r/d* = −8.5, but the height tends to the target, that is to say, with the increase of *Re* number, the rebound airflow in the stagnant zone is more restrained on the upstream wall jet, so it is difficult for it to escape from the upstream. In the downstream region, with the increase of *Re* number, the outflow of downstream air is increased. Although the radial position of entrainment vortex in the exit of jet space does not change significantly, the expansion scale increases slightly. For the SIJ, the low-speed recirculation vortex in the upstream region can be effectively eliminated. With the increase of *Re* number, the scale of the wall-like corner vortex formed at *r/d* = −2.5 in the stagnation zone decreases, so that the outer airflow entrained from the upstream region can participate in the downstream flow more smoothly. In the downstream region, the distribution of the flow field is basically the same as that of the circular hole jet. Under different *Re* numbers, a distinct recirculation vortex is formed near the exit region of the downstream jet space. However, with the increase of *Re*, the size of the vortex and the position of the center of the vortex do not change significantly.

### 3.2. Entropy Analysis

In this Section, entropy generation processes are analyzed to identify and quantify the causes of irreversibilities evolving in such inclined impinging cooling arrangements. As a second law of the thermodynamics analysis method, entropy production is be able to get the amount of irreversible loss and clarify the irreversible loss distribution in a flow and heat transfer system of concentrated irreversible phenomenon. Of course, entropy generation is an important optimization indicator to design a thermal system in terms of local entropy imbalance in the actual operation.

The volumetric entropy generation rate can be formulated as follows [33,34]:(5)$$S=\frac{k}{T^{2}}\left[{\left(\frac{\partial T}{\partial x}\right)}^{2}+{\left(\frac{\partial T}{\partial y}\right)}^{2}+{\left(\frac{\partial T}{\partial z}\right)}^{2}\right]$$

The right side of the equation is the volumetric entropy generation due to the heat transfer.

Figure 7 and Figure 8 shows the local entropy contour by heat transfer of the CIJ at different inclined angles on the Z_0_ plane when *H/d* = 2 and *Re* = 6000. The entropy production resulting from different flow configurations computed due to heat transfer is determined. In Figure 7, it appears that the entropy is predominantly produced in the regions in the vicinity of the impinged wall, that is, it may be the result of a large contribution of temperature transfer gradient and shear-induced turbulence mixing in these regions. The simulated results are consistent with those found in the literature. At the same time, the large entropy is the red areas corresponding to the regions near the jet axial outlet boundaries; this occurs because of the rapid temperature change across the jet axial outlet boundaries, that is, the temperature gradient is large in these region, which in turn results in a high entropy generation rate. For the same reason, the red area in the jet space corresponds to the flow vortex region, where the temperature gradient is relatively large. Under inclined conditions, entropy generation of CIJ and SIJ in the upstream is larger than that in the downstream and, with the increase in the inclined angle, the regions of large entropy generation extend. When α is equal to 75° and 90°, the region of large entropy generation is very small, while when α is smaller than 60°, the region of large entropy generation expands very rapidly. Compared with the CIJ, entropy generation of the SIJ in the upstream region is increased significantly as a result of more suction circumambient airflow, while that in the downstream region is weakened as a result of more homogeneous mixed with the airflow from the jet space. When α is equal to 45°, the whole region of entropy generation for the SIJ appears in the upstream region, the maximal entropy generation is the region of *r/d* = 4 corresponding to the wall jet region, and the vortex of entropy generation is the location of the wall-like corner vortex. It should be noted that reducing the inclined angle and use of the swirling flow can lead to a larger entropy generation, and the latter is the key measure. Therefore, the cooling/heating configuration with the SIJ will have a larger entropy generation under the same conditions compared with the CIJ. 

Figure 9 and Figure 10 show the local entropy contour by heat transfer of CIJ and SIJ at different impinging distances on the Z_0_ plane. In Figure 9, for a round hole jet, when *H/d* = 4, the region of entropy generation in the upstream direction is larger than that at *H/d* = 2. This is because increasing the impinging distance makes more air entrainment and flow mixing in the upstream direction, thus making the temperature gradient change more violent. When the impact distance increases further, the region of entropy generation decreases. In Figure 10, for the SIJ, the entropy generation region decreases with the increase of impinging distance.

Figure 11 and Figure 12 show the local entropy contour by heat transfer of the CIJ and SIJ at different Re on the Z_0_ plane. For the CIJ, increasing the jet Reynolds number has little effect on the region of entropy generation near the wall, while for the SIJ, the region of entropy generation in the upstream direction will gradually decrease, while there is almost no entropy generation in the downstream direction.

### 3.3. Local Heat Transfer Characteristics of Target Surface

Figure 13 shows the temperature contours of CIJ and SIJ at different inclination angles when *Re* = 6000 and *H/d* = 2. Whether it is CIJ or SIJ, reducing the inclination angle will reduce the uniformity of temperature distribution on the plate surface and there is a bigger area of high temperature in the stagnation zone. Figure 14 shows *Nu* number contours of CIJ and SIJ at different inclination angles when *Re* = 6000 and *H/d* = 2. For the vertical circular impinging jet, the stagnation zone is the maximum heat transfer region, and it is symmetrically distributed around the stagnation point near r = 0.5 d at a small jet-to-plate distance. At the same time, it can be seen that, with the decrease of the inclination angle of the jet, heat transfer performance on the downstream region of the target surface is enhanced, while the heat transfer performance in the upstream region of the target surface is seriously degraded, and the area of *Nu* peak region is also gradually reduced, that is to say, heat transfer uniformity of the whole target surface is deteriorated. Owing to the existence of four spiral grooves in the nozzle, four local *Nu* number peaks (140) are symmetrically distributed in the stagnation zone. When the jet is inclined, the two peak areas of *Nu* number in the downstream region disappear, and the other two peak areas in the upstream region extend into a larger area. At the same time, heat transfer performance can be maintained in the 1.5 d rectangular region of the peak region, which may be because of the rebound flow in the stagnation zone when the swirling jet is used. The maximum *Nu* in the upstream region of the stagnation zone also is degenerated to about 80 at 45°, and there are two secondary heat transfer peak areas on the upstream region of the stagnation zone, where there is also the region of the rebound airflow. Compared with the CIJ, the SIJ issuing from the new nozzle can greatly improve the heat transfer capacity in the stagnation zone of *r* = ±0.5 d, while maintaining a higher heat transfer performance in the rectangular region *r* = 1.5 d near the stagnation point. Even when the inclination angle is at a minimum, the peak value of *Nu* can reach heat transfer performance of the vertical circular impinging jet.

Figure 15 shows the variation curve of local *Nu* number on target surface of CIJ and SIJ at different inclination angles when *Re* = 6000 and *H/d* = 2. As can be seen from Figure 15, when the jet is vertical impingement, in the direction of *β* = 0°, the local *Nu* number at the stagnation point and the maximum *Nu* number for the SIJ increase by 98.5% and 114.6%, respectively, compared with the CIJ.

When *α* = 75°, it can be observed that, for CIJ and SIJ, the maximum *Nu* on the target surface is in the upstream region. In the range of 0 < *r/d* < 1.65 d, the *Nu* number in the upstream region for the CIJ is higher than that in the downstream region, while in the case of *r/d* > 1.65 d, the *Nu* number in the downstream region is higher than that in the upstream region; for the SIJ, the turning point of *Nu* number distribution in the upstream and downstream regions is at *r/d* = 0.9. At this time, local *Nu* number at the stagnation point and the maximum *Nu* number of the SIJ increased by 125.9% and 126.4%, respectively, compared with the circular jet. When the inclination angle is decreased from 90° to 75°, local *Nu* number distribution on the target surface does not change significantly for circular jet; while for the swirling jet, local *Nu* number at the stagnation point does not change significantly, but the maximum *Nu* number increases by 14.4%.

When *α* = 60°, the maximum *Nu* number of the target surface is still in the upstream region, whether it is a circular jet or a swirling jet. Compared with *α* = 75°, the turning point of *Nu* number distribution in the upstream and downstream regions for circular hole jet is slightly ahead, while it is at *r/d* = 0.75 for the swirling jet. At this time, the local *Nu* number at the stagnation point and the maximum Nu number for the SIJ increased by 139.9% and 116.3%, respectively, compared with the CIJ. It can also be seen that the radial position of local maximum *Nu* on the target surface of threaded and circular holes does not change significantly with the inclination of the jet from 90° to 60°. In terms of the local *Nu* value, the inclination does not weaken the heat transfer capacity near the stagnation zone on target surface of the CIJ and SIJ.

When *α* = 45°, the maximum *Nu* number of circular jet is still in the upstream region, and the turning point of *Nu* number distribution on the target surface is *r/d* = 1.1 *d*, while the *Nu* number in downstream direction of SIJ is always higher than that in the upstream direction. Compared with the CIJ, local *Nu* number at the center of the target surface and the maximum *Nu* number for the SIJ increased by 15.48% and 24.25%, respectively. Compared with the case of *α* = 60°, local *Nu* number at the stagnation point and the maximum *Nu* number of the circular hole jet at *α* = 45° have no obvious change, but they decrease by 46.8% and 53.3%, respectively, for the SIJ. It can be seen that, when the inclination angle is 45°, heat transfer effect on the CIJ is not obvious, but the heat transfer ability of the SIJ deteriorates significantly.

In the case of the inclined jet, the cooling effect of the SIJ is significantly stronger than that of the CIJ in the stagnant zone. Even when the inclination angle is 45°, the center of the heated surface and the maximum *Nu* number of the SIJ are still higher than those of the CIJ. The main reason for this phenomenon is that the central airflow of the swirling jet has a high exit velocity in jet space, which results in a strong convective heat transfer ability in the stagnation zone. When the inclination angle is 90°, 75°, and 60°, heat transfer enhancement effect of the SIJ is mainly concentrated in the region of *r/d* < 3 compared with the CIJ. When the airflow enters the region of wall jet flow, heat transfer effect of SIJ and CIJ is almost the same, and the *Nu* value is about 20. As far as the radial position of the maximum *Nu* is concerned, because the high-speed central area in the center of the threaded hole is relatively narrow, the local *Nu* peak value position is closer to the stagnation point than that of the circular hole. The maximum *Nu* number of circular hole jet is located at *r/d* = 0.5 when *α* = 90° and 75°, and at *r/d* = 0.7 when *α* = 60° and 45°. For a swirling impinging jet, the maximum *Nu* is at *r/d* = 0.25 when *α* = 90°, 75°, and 60°, and at *r/d* = 0.5 when *α* = 45°. By reducing the inclination angle of the jet, the positions of the maximum *Nu* number of the two jets are far away from the stagnation point of the target surface.

Figure 16 shows local Nusselt number distribution of CIJ and SIJ at different jet-to-plate distances at *Re* = 6000 and *α* = 45°. When *H/d* = 4, the maximum *Nu* number of SIJ and CIJ is in the downstream region. At this time, the maximum Nusselt number of the SIJ is 85.5% higher than that of the CIJ. When the jet-to-plate distance increases from *H/d* = 2 to *H/d* = 4, the local *Nu* number at the stagnation point and maximum *Nu* number decrease by 24% and 33.7%, respectively, while the *Nu* number at the stagnation point for the SIJ increases by 20.24%, and the maximum *Nu* number has no obvious change. When *H/d* = 6, the *Nu* in the downstream direction is still higher than in the upstream direction. At this time, the *Nu* number at the stagnation point decreases by 8.14%, and the maximum *Nu* number does not change significantly. In addition, when *H/d* = 6, the local *Nu* number at the stagnation point and the maximum *Nu* of the SIJ decreased by 54.85% and 45.6% compared with *H/d* = 4. It can be seen that the jet-to-plate distance increases from *H/d* = 2 to *H/d* = 4 when the inclination angle of the jet is 45°, the Nu number of the jet impingement core area (*r/d* < 1) decreases, but it does not change significantly when *H/d* = 6. For an SIJ, the *Nu* number at the stagnation point will increase when the jet-to-plate distance is increased to *H/d* = 4, but it will decrease sharply when *H/d* = 6. However, the maximum *Nu* of the SIJ is always higher than that of the CIJ in any jet-to-plate distance. Increasing the jet-to-plate distance causes the maximum *Nu* number of the CIJ to shift downstream, while that of the SIJ is opposite.

Figure 17a shows the distribution of local *Nu* number of SIJ and SIJ along the X-axis (*β* = 0°) at different Reynolds numbers when *H/d* = 2 and *α* = 45°. It can be seen from Figure 17a that increasing the Reynolds number, whether it is the circular impinging jet or swirling impinging jet, can significantly increase the local *Nu* number of the plate, but at a lower Reynolds number (*Re* = 6000), the local *Nu* distribution is more uniform. When the Reynolds number increases from 6000 to 24,000, the *Nu* number at the stagnation point increases by 27%, 90%, 138%, and 141%, respectively, compared with the CIJ. Figure 17b shows the maximum *Nu* of SIJ and SIJ at different Reynolds numbers when *H/d* = 2 and *α* = 45°. With the increase of Reynolds number, the difference between the maximum *Nu* number on the plate of SIJ and CIJ is gradually enlarged and tends to a certain value. When *Re* ≤ 18,000, increasing the Reynolds number makes the heat transfer effect of the SIJ very effective. When the Re is increased, most of mass flow is converged in the central region of the nozzle, and air flow along the helical grooves increases slightly with the hampered effect of the inclusion. So when the Re is larger than 18,000, the rate of enhanced heat transfer is gradually weakened.

### 3.4. Average Nusselt Number of the Target Surface (Nu_ave_)

Figure 18 shows the average *Nu* number of CIJ and SIJ at different inclination angles when *Re* = 6000 and *H/d* = 2. For the CIJ, the *Nu_ave_* at the inclined angle of 45° is 4.9%~23.56% lower than that at the other larger inclined angles, and the *Nu_ave_* distribution on the target surface is very close at the inclined angles of 60°, 75°, and 90°, which shows that, for CIJ, changing the inclined angle (α ≥ 60°) of the jet has little effect on the overall heat transfer of the target surface. In Figure 18b, when the inclination angle is 60°, 75°, and 90°, the *Nu_ave_* of the swirling impinging jet also changes slightly, but when the inclination angle is 45°, the *Nu_ave_* of the plate is much lower than that of the other jet inclination angles, and the decreased range is between 33% and 52.8%. Compared with the CIJ, the *Nu_ave_* of the SIJ is 38.5–95.6% higher when α = 90°, 38.7–109% higher when *α* = 75°, 34–106% higher when *α* = 60°, and 3.9–10.6% higher even when *α* = 45°. In conclusion, the *Nu_ave_* number distribution of the CIJ is more uniform than that of the SIJ, but the *Nu_ave_* of the SIJ is significantly higher than that of the CIJ. When the inclination angle is reduced to 45°, the heat transfer performance for CIJ and SIJ will be deteriorated significantly.

Figure 19 shows the *Nu_ave_* of CIJ and SIJ with different jet-to-plate distances at *Re* = 6000 and *α* = 45°. For the CIJ, the *Nu_ave_* at a jet-to-plate distance of *H/d* = 2 is 32.3%~52.4% higher than that at *H/d* = 4, but it does not change much when the jet-to-plate distance increases from *H/d* = 4 to *H/d* = 6. It can be seen that, when *Re* = 6000 and *α* = 45°, the heat transfer effect on the plate is better than when *H/d* = 2. For the *Nu_ave_* of the swirling impinging jet (Figure 19b), the *Nu_ave_* at *H/d* = 2 and 4 is higher than that at *H/d* = 6. By comparing and analyzing the *Nu_ave_* of swirling impinging jet with *H/d =* 2 and *H/d* = 4, it can be found that the *Nu_ave_* of *H/d* = 4 is higher than that at *H/d* = 2 at the region of *r/d* ≤ 1.3, but the opposite is true when r*/d* > 1.3. When the jet-to-plate distance is the same, the *Nu_ave_* of the swirling impinging jet is higher than that of the circular impinging jet. On the basis of the analysis of the distribution of the *Nu_ave_* and the local *Nu* numbers (Figure 16) of the SIJ at *Re* = 6000 and *α* = 45° under different jet-to-plate distances, it can be considered that the swirling jet has a better heat transfer effect on the target surface when *H/d* ≤ 4.

Figure 20 shows the distribution of the *Nu_ave_* of CIJ and SIJ at different Reynolds numbers when *H/d* = 2 and *α* = 45°. With the increase of Reynolds number, the *Nu_ave_* increases, and the increase of *Nu_ave_* of the SIJ is higher than that of the CIJ. For the CIJ, the *Nu_ave_* radial attenuation velocity does not change much at different Reynolds numbers, while for the SIJ, it increases gradually with the increase of Reynolds number. At the same Reynolds number, the *Nu_ave_* of the SIJ is higher than that of the CIJ, but considering the uniformity of heat transfer on the target surface, the Nusselt number distribution of the CIJ is more uniform.

## 4. Conclusions

In this paper, the effects of inclination angle, jet-to-plate distance, and Reynolds number on the flow and heat transfer in the swirling and non-swirling inclined impinging jets are studied. The main results are as follows:(1)By comparing with experimental data, it is verified that the SST *k-w* model can simulate the swirling impinging jet well under inclined conditions. Inclination angle and jet-to-plate distance have a significant influence on flow field, while the Reynolds number has a slight effect on flow structure. This is showed that there is a huge difference of flow structures in jet space between the CIJ and the SIJ under the same conditions. At a low jet spacing of *H*/*d* = 2, Reducing the inclination angle will make the recirculation vortex central in the upstream region to move towards the stagnation point of the plate for CIJ and SIJ. At the same, the scale of the recirculation vortex in the upstream region will gradually increase for the CIJ, while this recirculation vortex is gradually weakened, and eventually becomes a very small wall-like corner vortex in the stagnation zone for the SIJ. When *H/d* is bigger than 4, the large scale of recirculation vortex in the upstream region for CIJ and SIJ disappears, while the vortex near the downstream exit occurs for the CIJ and there are no any vortices in the downstream region for the SIJ.(2)No matter for the CIJ or SIJ, reducing the inclination angle will enlarge the entropy generation area, because reducing the inclination angle will enhance the entrainment effect of the jet on the external air, thus increasing the flow mixing and making the temperature gradient change greatly. Under the same conditions, SIJ will have a larger entropy generation under the same conditions compared with the CIJ. When the impingement distance increases, the entropy generation region in the upstream direction of the CIJ increases first and then decreases, while the entropy generation region of the SIJ decreases gradually. The change of Reynolds number has little effect on the entropy generation region of the round hole jet, but it will reduce the entropy generation region in the upstream direction of the SIJ, and there is almost no entropy generation in the downstream direction.(3)When the Reynolds number and jet-to-plate distance are constant, the maximum *Nu* number point of CIJ is always located in the upstream direction of the plate, and the change of *Nu* is not obvious. For the SIJ, when the inclination angle decreases to 45°, the maximum *Nu* point will shift from upstream to downstream. When the inclination angle varies between 60° and 90°, the *Nu* of swirling impinging jet changes little, but when the inclination angle is 45, the *Nu* decreases obviously. In the case of the inclined jet, the cooling effect of the SIJ is significantly stronger than that of the CIJ in the stagnant zone. Even when the inclination angle is 45°, the Nu at the stagnation point and the maximum Nu number of the SIJ are still higher than those of the CIJ.(4)When the Reynolds number and the inclination angle of the jet are constant, the *Nu* number in the stagnation zone (*r/d* < 1) decreases gradually with the increase of the jet-to-plate distance *H*. When the jet-to-plate distance is *H/d* > 2, the *Nu_ave_* of circular impinging jet will decrease. For the swirling impinging jet, with the increase of jet-to-plate distance from 2 d to 6 d, the *Nu* in the stagnation zone (*r/d* < 1) first increases and then decreases, while the *Nu_ave_* does not differ much until *H/d* = 6. In addition, when the jet-to-plate distance is constant, the *Nu_ave_* of swirling impinging jet is higher than that of circular impinging jet.(5)Under inclined conditions, when the jet-to-plate distance is small, the *Nu* of both nozzles increases with the increase of the Reynolds number. When the Re is increased, most of mass flow is converged in the central region of the nozzle and air flow along the helical grooves increases slightly with the hampered effect of the inclusion. So when the Re is larger than 18,000, the rate of enhanced heat transfer gradually weakens.

## Figures and Tables

**Figure 1 entropy-22-00015-f001:**
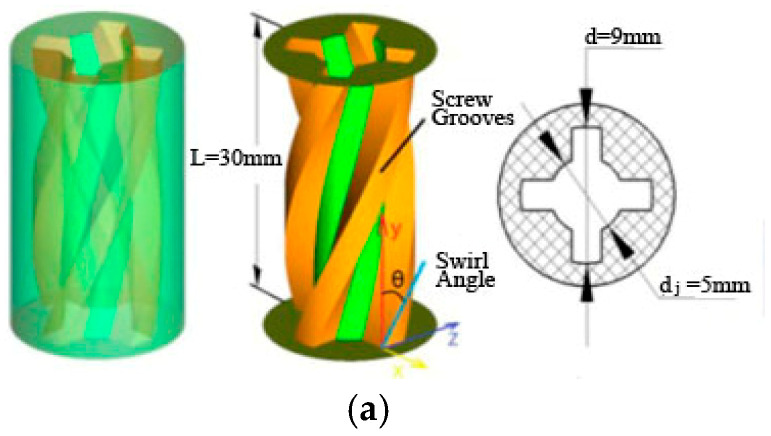
Calculation model: (**a**) the hole with a thread-like structure; (**b**) computational domain of the inclined swirling impinging jet (SIJ).

**Figure 2 entropy-22-00015-f002:**
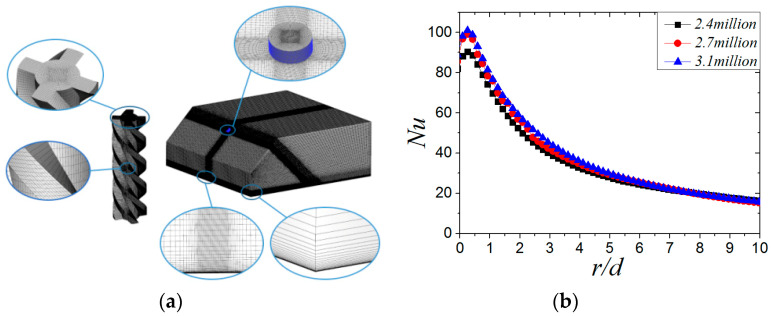
Grid and grid independence test: (**a**) multi-block structured grid; (**b**) swirling impinging jet (SIJ), α = 45°, *H/d* = four grid independence tests.

**Figure 3 entropy-22-00015-f003:**
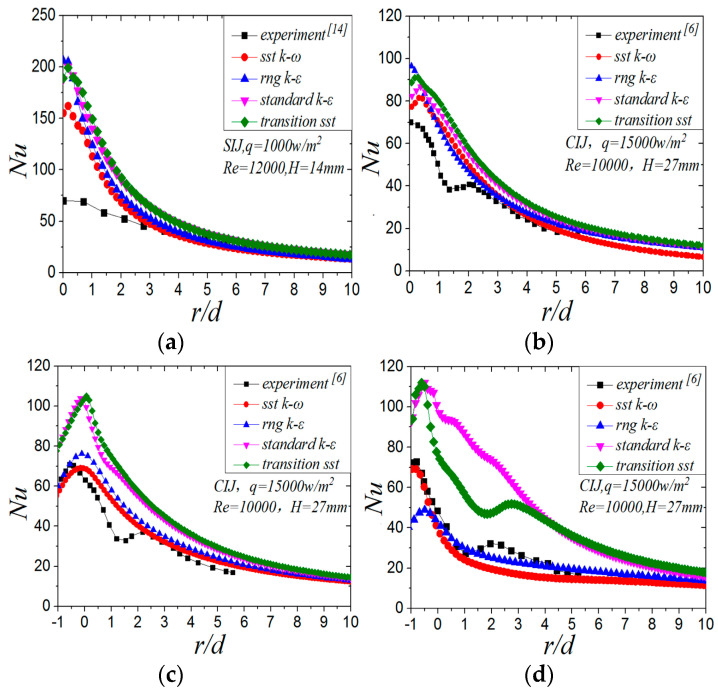
*Nu* number of target surface of different turbulence models: (**a**) *α* = 90°*，β* = 0°, *d* = 7 mm*,* SIJ; (**b**) *α* = 90°, *β* = 0°, *d* = 13.5 mm, circular impinging jet (CIJ); (**c**) *α* = 75°*, β* = 0°, *d* = 13.5 mm*,* CIJ; (**d**) *α* = 60°*, β* = 0°, *d* = 13.5 mm, CIJ.

**Figure 4 entropy-22-00015-f004:**
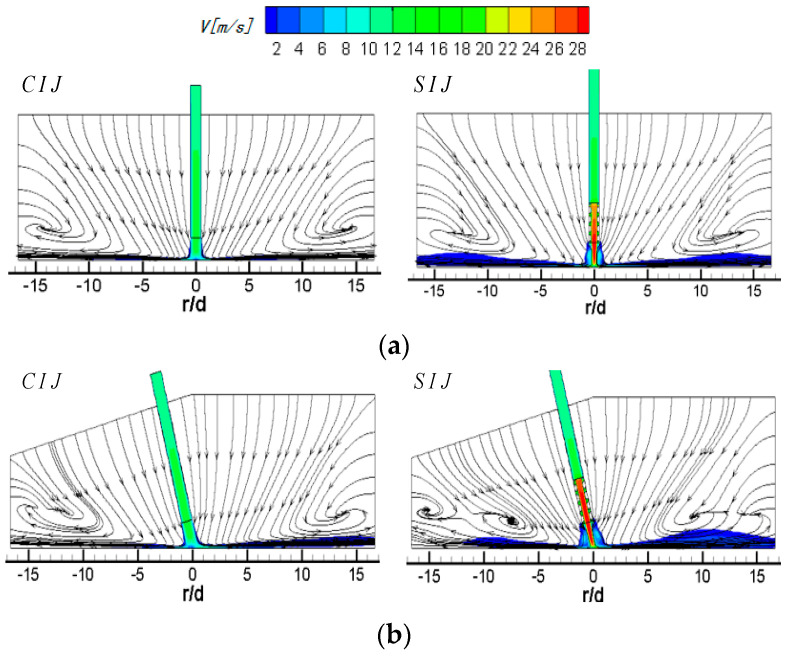
Streamlines and velocity contours of circular impinging jet (CIJ) and swirling impinging jet (SIJ) at different inclined angles on the Z_0_ plane: (**a**) *α* = 90°; (**b**) *α* = 75°; (**c**) *α* = 60°; (**d**) *α* = 45°.

**Figure 5 entropy-22-00015-f005:**
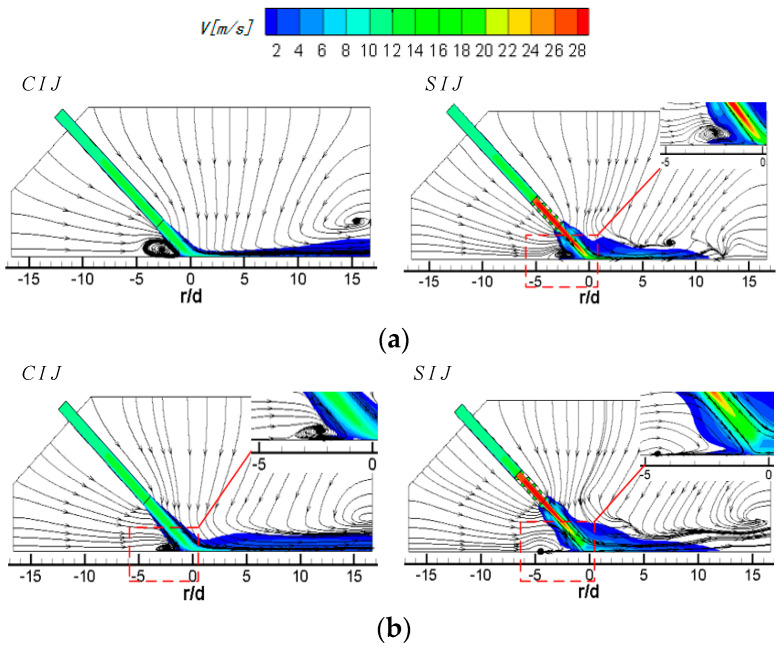
*Re* = 6000, *α* = 45°; streamlines and velocity contours of CIJ and SIJ under different jet-to-plate distances: (**a**) *H/d* = 4; (**b**) *H/d* = 6.

**Figure 6 entropy-22-00015-f006:**
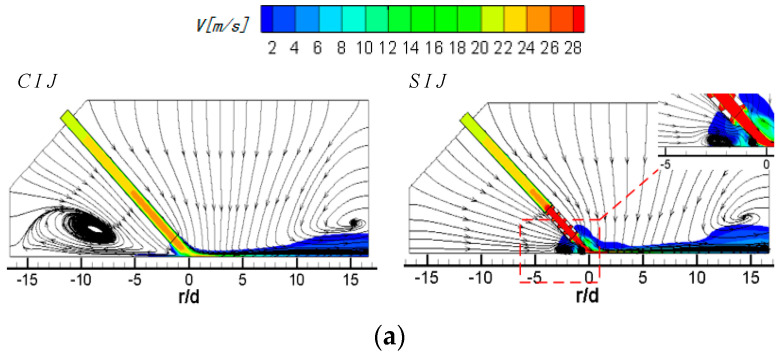
*H/d* = 2, *α* = 45°; streamlines and velocity contours of the CIJ and SIJ at different Reynolds numbers: (**a**) *Re* = 12,000; (**b**) *Re* = 18,000; (**c**) *Re* = 24,000.

**Figure 7 entropy-22-00015-f007:**
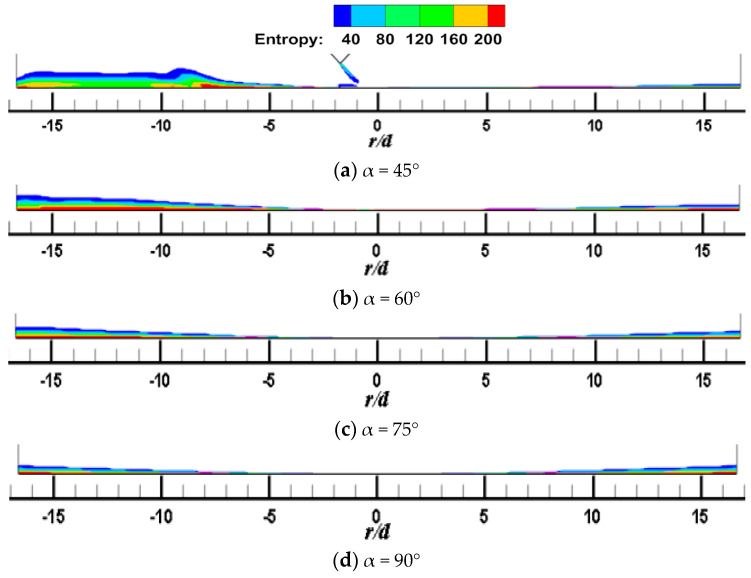
*H/d* = 2, *Re* = 6000; local entropy contour by heat transfer of the CIJ at different inclined angles on the *Z*_0_ plane.

**Figure 8 entropy-22-00015-f008:**
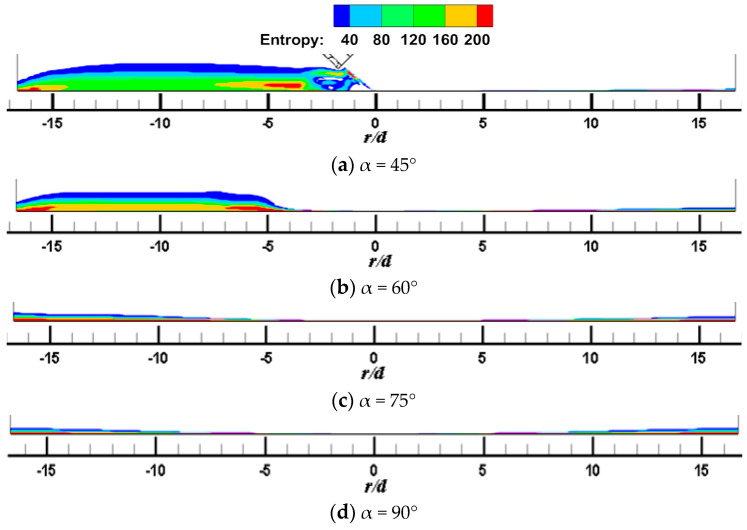
*H/d* = 2, *Re* = 6000; local entropy contour by heat transfer of the SIJ at different inclined angles on the Z_0_ plane.

**Figure 9 entropy-22-00015-f009:**
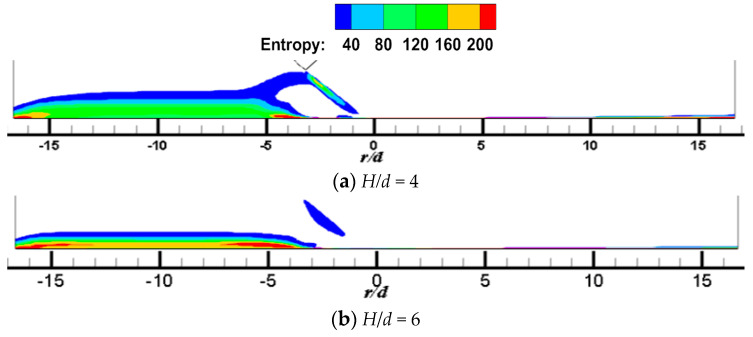
*α* = 45°, *Re* = 6000; local entropy contour by heat transfer of the CIJ at different jet distance on the *Z*_0_ plane: (**a**) *H*/*d* = 4, (**b**) *H*/*d* = 6.

**Figure 10 entropy-22-00015-f010:**
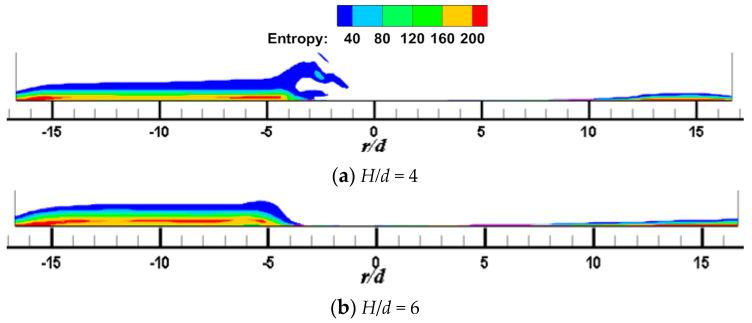
*α* = 45°, *Re* = 6000; local entropy contour by heat transfer of the SIJ at different jet distances on the *Z*_0_ plane: (**a**) *H/d* = 4, (**b**) *H/d* = 6.

**Figure 11 entropy-22-00015-f011:**
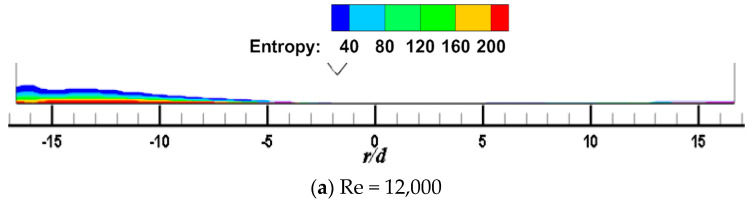
*α* = 45°, *Re* = 6000; local entropy contour by heat transfer of the CIJ at different *Re* on the *Z*_0_ plane: (**a**) *Re* = 12,000, (**b**) *Re* = 18,000, (**c**) *Re* = 24,000.

**Figure 12 entropy-22-00015-f012:**
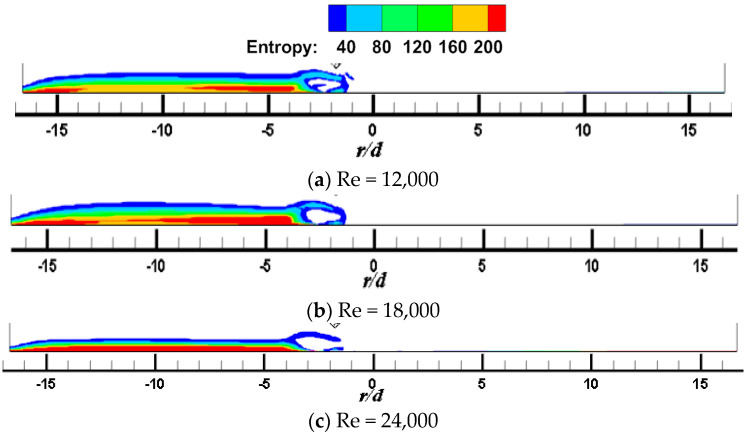
*α* = 45°, *Re* = 6000; local entropy contour by heat transfer of the SIJ at different Re on the *Z*_0_ plane: (**a**) *Re* = 12,000, (**b**) *Re* = 18,000, (**c**) *Re* = 24,000.

**Figure 13 entropy-22-00015-f013:**
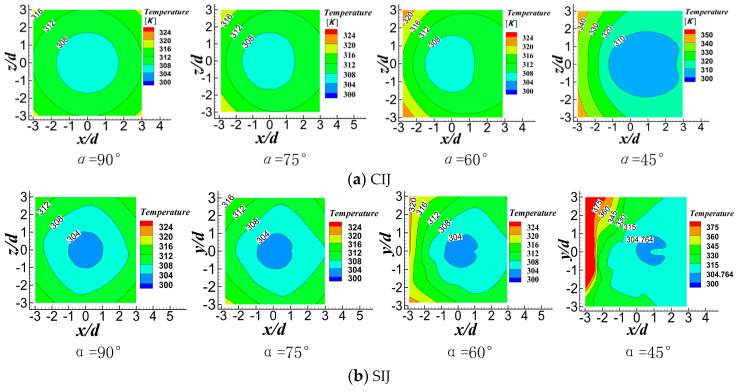
Temperature distribution of target surface at different inclination angles at *Re* = 6000 and *H/d* = 2: (**a**) CIJ, (**b**) SIJ.

**Figure 14 entropy-22-00015-f014:**
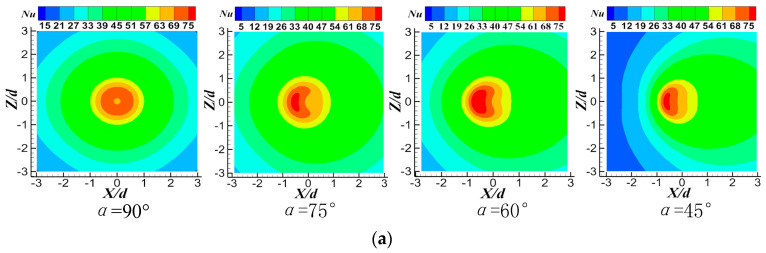
Nusselt number distribution of the target surface at different inclination angles at *Re* = 6000 and *H/d* = 2: (**a**) CIJ, (**b**) SIJ.

**Figure 15 entropy-22-00015-f015:**
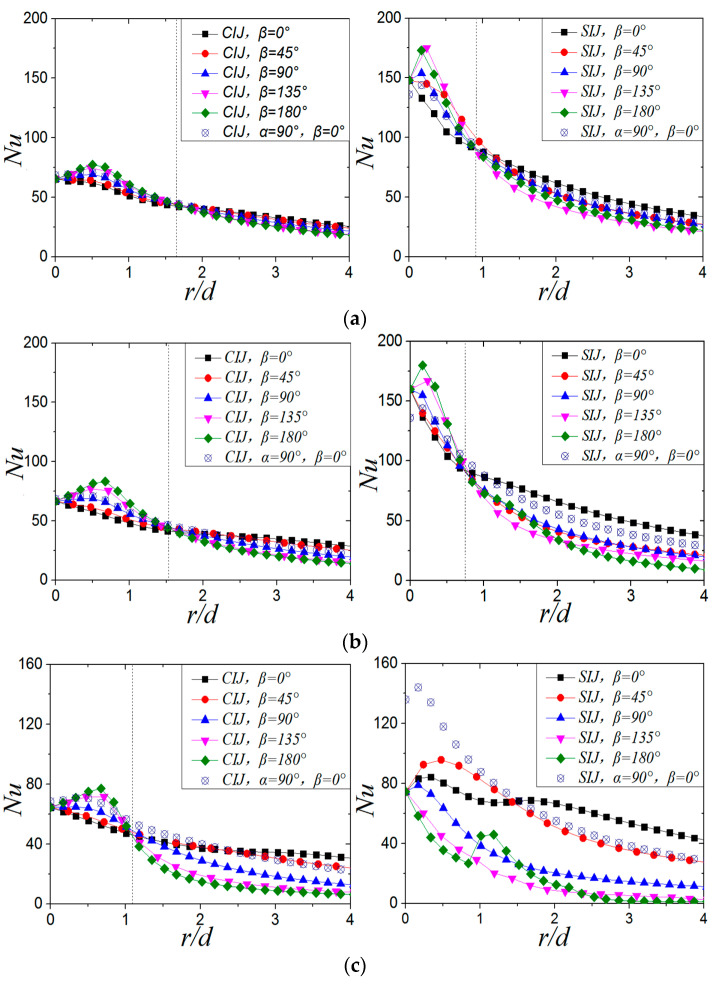
Nusselt number distribution on target surface at different jet inclination angles. (**a**) *α* = 75°, (**b**) *α* = 60°, (**c**) *α* = 45°.

**Figure 16 entropy-22-00015-f016:**
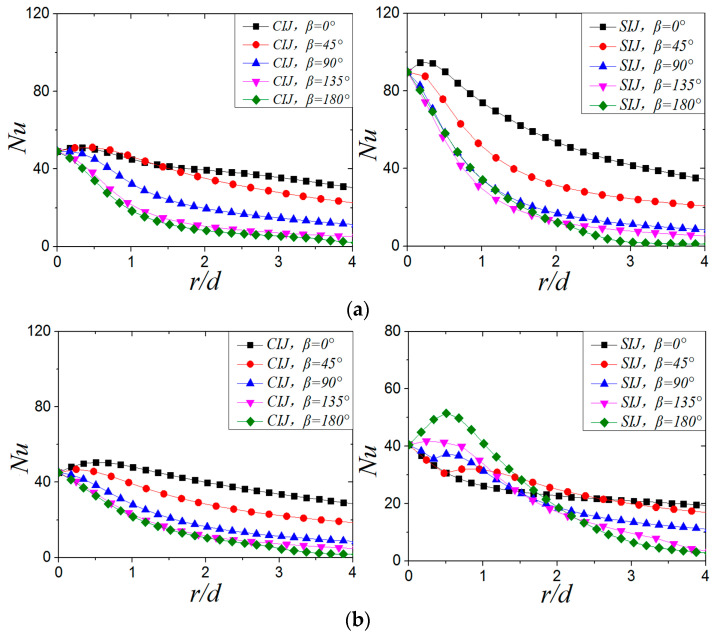
*Re* = 6000, *α* = 45°; local Nu number distribution on target surface at different jet jet-to-plate distances: (**a**) *H/d* = 4, (**b**) *H/d* = 6.

**Figure 17 entropy-22-00015-f017:**
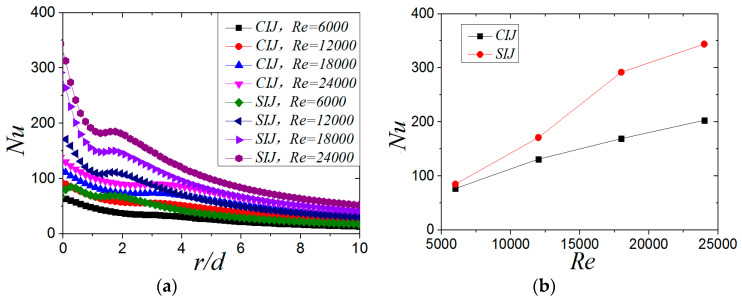
(**a**) *α* = 45° and *β* = 0°; the local Nu number distribution. (**b**) *H/d* = 2, *α* = 45°; the maximum *Nu* number of CIJ and SIJ at different Reynolds numbers.

**Figure 18 entropy-22-00015-f018:**
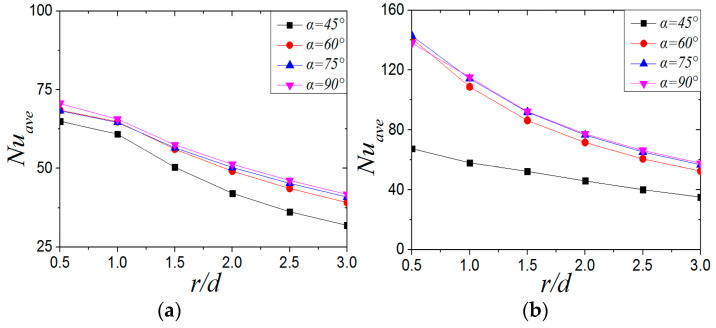
*Re* = 6000, *H/d* = 2; *Nu_ave_* number distribution of plates at different tilt angles: (**a**) CIJ, (**b**) SIJ.

**Figure 19 entropy-22-00015-f019:**
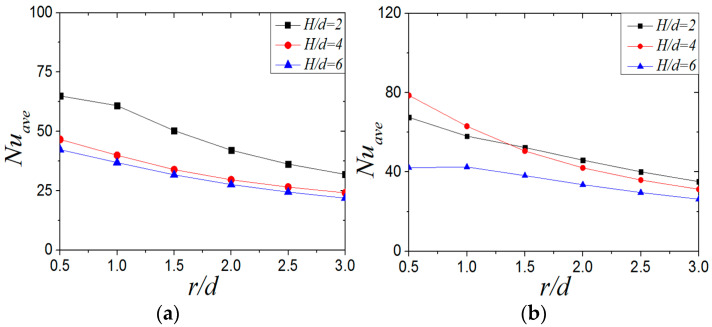
*Re* = 6000, α= 45°; *Nu_ave_* number distribution of plate surface under different jet-to-plate distances: (**a**) CIJ, (**b**) SIJ.

**Figure 20 entropy-22-00015-f020:**
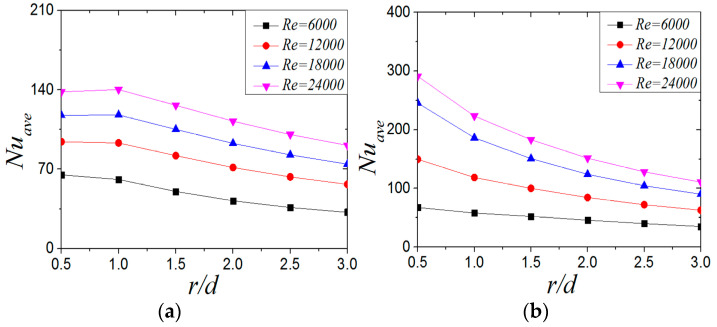
The distribution of average *Nu_ave_* number of plates under different Reynolds numbers: (**a**) CIJ, (**b**) SIJ.

**Table 1 entropy-22-00015-t001:** Conditions of operation studied in this paper. CIJ, circular impinging jet; SIJ, swirling impinging jet.

Parameter	Value
	CIJ/SIJ
*α*	45°, 60°, 75°, 90°
*H/d*	2, 4, 6
*Re*	6000, 12,000, 18,000, 24,000
*q* (W/m^2^)	1000

**Table 2 entropy-22-00015-t002:** Massflow rate and pressure.

*Re*	Mass Flow Rate (kg/s)	Inlet Pressure (pa)	Outlet Pressure (pa)
6000	7.833 × 10^−4^	101,786.4	101,325
12000	1.567 × 10^−3^	102,922.8	
18000	2.35 × 10^−3^	104,701.6	
24000	3.133 × 10^−3^	107,121.52	

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
