# Peer review of "Numerical Simulation of Swirling Impinging Jet Issuing from a Threaded Hole under Inclined Condition"

_entropy, 2019, doi:10.3390/e22010015_

Round 1

Reviewer 1 Report

One of my general comments concerns the "matching" of the article to the subject of the journal. The word entropy or optimization does not appear in the article. With such a research tool used, it is worth linking it to some energy parameter, it would be ideal to use entropy or exergy for this purpose. In the text, a lot of typing mistakes can be founded. General English has to be improved. Sometimes there is the problem with understanding the content. For the turbulence flow, it is very important to inform the readers about y+ function. Why this domain was chosen? What kind of boundary conditions were used? It has to clarify in the text. Did you study the influence of boundary conditions on the simulation? Page no 4 is w/m^2 should be W/m^2 Is w/m*K should be W/m*K Which version of Fluent was used during the calculation? I think it can be interested to show the temperature contours within the text. The conclusion has to be modified, some comments are too obvious.

Author Response

Dear Editors and Reviewers: Thank you for your letter and the reviewers’ comments concerning our manuscript entitled “Numerical Simulation of Swirling Impinging Jet issuing from a Threaded Hole under Inclined Condition”. Those comments are all valuable and very helpful for revising and improving our paper, have important guiding significance to our research. We have studied the comments carefully and have revised the manuscript according to them. We hope we can meet with approval. Revised portion are marked in blue. The main revision and the responses to the comments are as follows: Responses to the comments: Reviewer #1:

Comments and Suggestions for Authors:One of my general comments concerns the "matching" of the article to the subject of the journal. The word entropy or optimization does not appear in the article. With such a research tool used, it is worth linking it to some energy parameter, it would be ideal to use entropy or exergy for this purpose. In the text, a lot of typing mistakes can be founded. General English has to be improved. Sometimes there is the problem with understanding the content. For the turbulence flow, it is very important to inform the readers about y+ function. Why this domain was chosen? What kind of boundary conditions were used? It has to clarify in the text. Did you study the influence of boundary conditions on the simulation? Page no 4 is w/m^2 should be W/m^2 Is w/m*K should be W/m*K .Which version of Fluent was used during the calculation? I think it can be interested to show the temperature contours within the text. The conclusion has to be modified, some comments are too obvious. 

1.One of my general comments concerns the "matching" of the article to the subject of the journal. The word entropy or optimization does not appear in the article. With such a research tool used, it is worth linking it to some energy parameter, it would be ideal to use entropy or exergy for this purpose.

Reply:Thank you very much for your advice, and we have added the descriptions of entropy contour in Section 3.1.

2.In the text, a lot of typing mistakes can be founded. General English has to be improved. Sometimes there is the problem with understanding the content.

Reply:Thank you very much for your advice, and we have checked this paper carefully.

3.For the turbulence flow, it is very important to inform the readers about y+ function. Why this domain was chosen?

Reply:In order to capture the flow field near the wall accurately, the value of y+ function should be less than 1 when the SST K-W turbulence model is selected. So the interface region of the jet flow space above the target surface is chosen as a focus domain to use a structured and refined grid. In this paper, the height of the first layer of this domain is 0.001mm with the grid growth rate of 1.2, and then the value of y+ function is about 0.8.

4.What kind of boundary conditions were used? It has to clarify in the text. Did you study the influence of boundary conditions on the simulation?

Reply: Thank you very much for your advice, and we have added detailed descriptions of boundary conditions. The influence of boundary conditions was not studied.

5.Page no 4 is w/m^2 should be W/m^2 Is w/m*K should be W/m*K .Which version of Fluent was used during the calculation?

Reply:The unit in the article has been modified correctly. In this paper, ANSYS fluent 18.0 is used for calculation.

6.I think it can be interested to show the temperature contours within the text.

Reply: Thank you very much for your advice, and we have added temperature contours on the target surface in Section 3.2. 7.The conclusion has to be modified, some comments are too obvious.  Reply:Thank you very much for your advice, and we have modified the conclusion.

Reviewer 2 Report

In this work, XU Liang et al., investigated the heat transfer performance of a swirling gas jet impinging on an inclined metallic surface. It is shown numerically that by introduction of angular momentum to the jet, the cooling efficiency can be significantly improved. The impact of system parameters such as jet to plate distance, various tilt angles and Reynolds number are examined.

I think that the results are interesting and warrant publication in Entropy. However, there are some minor issues that need clarification.

1. The article is a continuation of the work [14], where cooling efficiency has been examined in the case of perpendicular jet direction. I couldn't find an english version of the manuscript, but from the abstract and figures one can conclude that in the perpendicular case, the efficiency increases with nozzle helix angle, e. g. large angular momentum is beneficial. Does this hold in the case of inclined surface too? Is there some optimum twist angle? The angles of up to 45 degrees were tested, and the largest 45 degree twist nozzle has been selected for this work. What happens at even larger twist angle? I understand that twist angle optimisation is beyond the scope of the manuscript, but some comment on this would be helpful.

2. The authors state that the entrance boundary condition is a constant mass flow inlet. What is this mass flow exactly? This probably could be figured out from geometry and Reynolds number, but explicit numbers may be helpful for the reader. Moreover, in practical applications the inlet may provide some constant pressure instead of constant mass flow. Under these conditions, the mass flow will be hampered by the inclusion of helical grooves. Would there still be any cooling advantage then?

3. Table 1 is divided between CIJ/SIJ, but all parameters below are common to both cases. I believe that division is not necessary.

4. At the end of page 12, the authors state that an increase of Reynolds number beyond Re=18000 does not increase the heat transfer significantly. This is also repeated in conclusions. Any comments on why this happens?

Author Response

Dear Editors and Reviewers:

Thank you for your letter and the reviewers’ comments concerning our manuscript entitled “Numerical Simulation of Swirling Impinging Jet issuing from a Threaded Hole under Inclined Condition”. Those comments are all valuable and very helpful for revising and improving our paper, have important guiding significance to our research. We have studied the comments carefully and have revised the manuscript according to them. We hope we can meet with approval. Revised portion are marked in blue. The main revision and the responses to the comments are as follows:

Responses to the comments:

Reviewer #2:

Comments and Suggestions for Authors:In this work, XU Liang et al., investigated the heat transfer performance of a swirling gas jet impinging on an inclined metallic surface. It is shown numerically that by introduction of angular momentum to the jet, the cooling efficiency can be significantly improved. The impact of system parameters such as jet to plate distance, various tilt angles and Reynolds number are examined. I think that the results are interesting and warrant publication in Entropy. However, there are some minor issues that need clarification.

1.The article is a continuation of the work [14], where cooling efficiency has been examined in the case of perpendicular jet direction. I couldn't find an english version of the manuscript, but from the abstract and figures one can conclude that in the perpendicular case, the efficiency increases with nozzle helix angle, e. g. large angular momentum is beneficial. Does this hold in the case of inclined surface too? Is there some optimum twist angle? The angles of up to 45 degrees were tested, and the largest 45 degree twist nozzle has been selected for this work. What happens at even larger twist angle? I understand that twist angle optimisation is beyond the scope of the manuscript, but some comment on this would be helpful.

Reply: Thank you very much for your advice, helix angle of 45° is chosen as a representative to study the flow and heat transfer characteristics of the swirling impinging jet under the inclined conditions. Our presented nozzle is similar to the conventional circular hole with an internal threads. Flow regime at the nozzle exit is up to the nozzle structures and inlet airflow conditions, so the optimum twist angle for the vertical impinging jet may be hold in the case of inclined condition.

The optimum twist angle is an interesting topic. With the increase of helix angle, the airflow in the four circumferential screw grooves will be decreased by the higher flow resistance, and then most of airflow is just ejected along the hole central with a low angular momentum. From this point of view, there is some optimum helix angle. In the future, we will study channel number、helix angle、filled radius of the threads to explore the optimum swirler. Thanks you again. we have added the results of different helix angles for the vertical impinging jet in Section 1.

 2.The authors state that the entrance boundary condition is a constant mass flow inlet. What is this mass flow exactly? This probably could be figured out from geometry and Reynolds number, but explicit numbers may be helpful for the reader. Moreover, in practical applications the inlet may provide some constant pressure instead of constant mass flow. Under these conditions, the mass flow will be hampered by the inclusion of helical grooves. Would there still be any cooling advantage then?

Reply: Thank you very much for your advice, we have added mass flow and inlet total pressure in Table 2. The inclusion of helical grooves will surely hamper inflow, it should be a design parameter in the actual application. While the nozzle length of impinging jet is generally short for the application of gas turbine cooling blades and electronic components, pressure loss may  be not a big significant increase.

3.Table 1 is divided between CIJ/SIJ, but all parameters below are common to both cases. I believe that division is not necessary.

Reply: Thank you very much for your advice, we have modified it in Table 1.

4.At the end of page 12, the authors state that an increase of Reynolds number beyond Re=18000 does not increase the heat transfer significantly. This is also repeated in conclusions. Any comments on why this happens?

Reply: Under the inclined conditions, the heat transfer coefficient on the target surface for the swirling jet is increased totally with the increasing of the Re. When the Re is increased, most of mass flow is converged in the central region of the nozzle and air flow along the helical grooves increases slightly with the hampered effect of the inclusion. So when the Re is larger than 18000, the rate of enhanced heat transfer is gradually weakening.

Thank you very much for your advice, we have added it.

Round 2

Reviewer 2 Report

The Authors have made a significant effort to improve the manuscript and answered all my questions. The new section on entropy analysis is interesting and fits well in the journal scope. I suggest publication in the current form.